# Factors associated with accessing and utilisation of healthcare and provision of health services for residents of slums in low and middle-income countries: a scoping review of recent literature

Ji-Eun Park [1,2] Peter Kibe [3] Godwin Yeboah [4] Oyinlola Oyebode,[1] Bronwyn Harris [1] Motunrayo M Ajisola,[5] Frances Griffiths [1,6] Navneet Aujla,[1,7] Paramjit Gill [1] Richard J Lilford [8] Yen-Fu Chen [1] The Improving Health in Slums Collaborative

For numbered affiliations see end of article.

**Correspondence to**
Dr Yen-Fu Chen;
y-f.chen@warwick.ac.uk

## ABSTRACT

**Objective** To identify factors associated with accessing and utilisation of healthcare and provision of health services in slums.

**Design** A scoping review incorporating a conceptual framework for configuring reported factors.

**Data sources** MEDLINE, Embase, CINAHL, Web of Science and the Cochrane Library were searched from their inception to December 2021 using slum-related terms.

**Eligibility criteria** Empirical studies of all designs reporting relevant factors in slums in low and middle-income countries.

**Data extraction and synthesis** Studies were categorised and data were charted according to a preliminary conceptual framework refined by emerging findings. Results were tabulated and narratively summarised.

**Results** Of the 15 469 records retrieved from all years, 4368 records dated between 2016 and 2021 were screened by two independent reviewers and 111 studies were included. The majority (63 studies, 57%) were conducted in Asia, predominantly in India. In total, 104 studies examined healthcare access and utilisation from slum residents' perspective while only 10 studies explored provision of health services from providers/planners' perspective (three studies included both). A multitude of factors are associated with accessing, using and providing healthcare in slums, including recent migration to slums; knowledge, perception and past experience of illness, healthcare needs and health services; financial constraint and competing priorities between health and making a living; lacking social support; unfavourable physical environment and locality; sociocultural expectations and stigma; lack of official recognition; and existing problems in the health system.

**Conclusion** The scoping review identified a significant body of recent literature reporting factors associated with accessing, utilisation and provision of healthcare services in slums. We classified the diverse factors under seven broad categories. The findings can inform a holistic approach to improving health services in slums by tackling barriers at different levels, taking

### Strengths and limitations of this study

⇒ We conducted literature search in multiple databases using generic terms related to slums to ensure that a wide range of relevant studies was captured.

⇒ A conceptual framework explaining factors associated with accessing and utilisation of healthcare by slum residents as well as provision of healthcare in slums was developed and used to categorise identified studies and factors.

⇒ We examined barriers and facilitators of accessing healthcare and service provision from the perspectives of both demand side (slum residents) and supply side (healthcare providers and service planners).

⇒ Only studies published in academic journals between 2016 and 2021 in English language were included, and methodological quality of each included study was not examined because of time constraint.

⇒ We did not explore the complex relationships and interactions between various factors in different contexts at different slum locations, but our mapping of these factors to the conceptual framework should facilitate further in-depth analyses.

into account local context and geospatial features of individual slums.

**Systematic review registration number** https://osf.io/694t2.

## INTRODUCTION

Rapid urbanisation has resulted in a growing number of residents in slums[1] who face ongoing problems such as unemployment, poor sanitation, lack of transport, high level of crime and haphazard development.[2] In 2018, over 1 billion people were living in slum-like conditions, and Central, South and South-East Asia and sub-Saharan Africa



accounted for 80% of them.[1] Even though various definitions of slums exist, there is no universally agreed definition of what constitutes 'a slum', and the term itself is widely debated and contested.[3 4] For the purpose of this scoping review, we refer to slums as densely populated areas characterised by lack of basic services, substandard housing, overcrowding, unhealthy living condition, insecure tenure and poverty,[4 5] taking into account the crucial concepts of place and space that are important in shaping health outcomes and community access to health services in these urban settings.[4]

Previous studies have reported various risk factors affecting health of slum residents such as physical environment,[6] sanitation,[7] social capital[8 9] and water governance,[10] and have observed in some cases that slum residents have worse health status compared with non-slum urban and/or rural residents. For example, Ezeh *et al* found that children living in slums had higher mortality than rural and non-slum urban populations.[3] Poorer height for age for children[11] and higher prevalence of childhood illnesses and malnutrition[12] have also been observed in slums compared with non-slum urban and rural settings. In addition, slum residents are susceptible to unhealthy behaviours.[13 14] Living in slums has been found to be associated with low physical activity,[13] poor diet[14] and poor knowledge about the cause and preventability of diseases.[15]

Despite the unfavourable health status and environment, and consequently the potential high level of healthcare needs, previous studies showed that slum residents were less likely to seek and use healthcare services than their non-slum counterparts in the cities.[16 17] Slum residents have been found to have lower rates of healthcare utilisation in antenatal services[16] and services for non-communicable diseases[17] compared with residents of urban 'formal' settings. One study in Iran showed that only about half of slum households that required outpatient services could use them.[18] Another study in Haiti also reported that one-third of slum households were not able to access medical care for their children when it was needed in the past year.[19]

While the health status and needs of slum residents have been described in previous reviews,[3 20] factors associated with healthcare-seeking behaviour and healthcare utilisation of slum residents and factors related to the provision of health services in slums have not been systematically examined (with the exception of immunisation services).[21] This scoping review aims to fill in these evidence gaps and inform efforts to improve healthcare delivery to people in slums.

## METHODS

This scoping review was performed according to current best practice guidance.[22] The broad question of interest was: '*What factors are associated with slum residents' accessing and utilisation of health care and/or the provision of health services in slum settings in low and middle income countries*

*(LMICs)*?'. The protocol for this review was registered in Open Science Framework.[23]

### Literature search and study selection

A broad search of five databases (MEDLINE, Embase, CINAHL, Web of Science and the Cochrane Library) was conducted in April 2020 and updated in December 2021. Searches were limited to English language. Key terms related to slums were used: slum or slums or ghetto or ghettos or informal settlement$ or shantytown$ or shanty town$ or favela$ (online supplemental appendix 1). We did not include terms related to other concepts in order to maximise the sensitivity of our searches. In addition, we searched the organisational websites of Slum Dwellers International, UN HABITAT, United Nations and WHO but did not identify relevant studies.[24–27]

Records retrieved from databases (after duplicates were removed) were initially screened by one reviewer (J-EP) and those which did not meet the inclusion criteria were disregarded. After that, a second reviewer (PK, GY, OO) examined the remaining records independently based on titles and abstracts. When the decisions of two reviewers differed, the discrepancy was resolved based on full texts and/or by discussion with a third reviewer (Y-FC) or the broader review team. This study-screening process started from records of the most recent years (ie, in the past 3 years) and then proceeded to prior years. Due to the larger than expected volume of the literature, we eventually screened records between 2016 and 2021 and did not cover earlier records in order to synthesise and present the findings from latest evidence in a timely fashion to inform the wider project hosting this review.[28 29]

### Inclusion and exclusion criteria

A study was included when it: (1) described factors related to slum residents' accessing or utilisation of healthcare or the provision of health services in slums; and (2) was conducted in relation to slums in LMICs. Only articles written in English were included. A study was excluded when it was a commentary, opinion or narrative review; described slum residents' utilisation of health services or the provision of health services without exploring the associated factors; investigated informal care at home; or included mixed slum and non-slum populations without separately reporting data for slum residents or investigating residency in slums as a factor for healthcare access.

During our updated search in December 2021, we found several studies reporting healthcare utilisation[28] and provision related to COVID-19 in slums.[30] These studies were not included in this scoping review, since the factors associated with healthcare utilisation and health service provision under the pandemic situation are dramatically different and warrant a separate synthesis.

We included both primary studies and systematic reviews that examine data collected empirically and that derive their findings based on the data. Both quantitative and qualitative studies (and by extension, mixed methods studies) were considered. Even though slums have existed

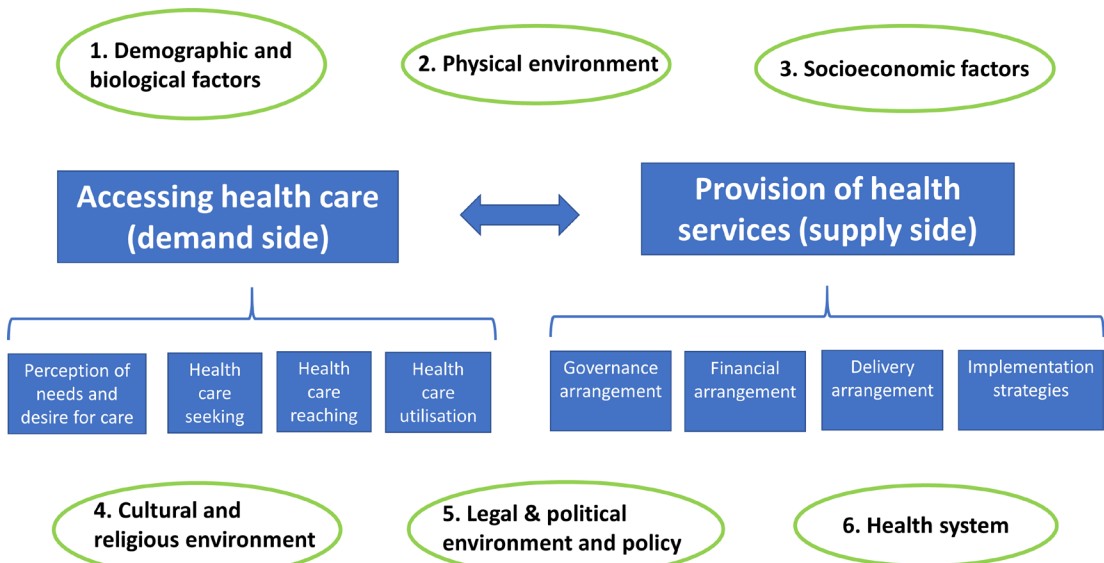

**Figure 1** Preliminary framework for factors influencing slum residents' healthcare-seeking behaviour and utilisation of health services and the provision of services in slum settings.

in both high-income countries and LMICs, the context may be quite different between these countries. For example, while all slums are vulnerable to natural disasters such as tropical cyclones, the impact of these could be far more severe in slums of LMICs due to the different socioeconomic contexts.[31] In this review, we focused on settings in LMICs and excluded studies conducted in high-income countries.

### Study coding and data extraction/charting

Eligible studies were coded and data extracted/charted according to a prespecified preliminary framework shown in figure 1. The preliminary framework was developed by the review authors based on existing conceptual models

related to healthcare access and service delivery[32–35] and was modified during the scoping review process to accommodate new factors/themes identified from the literature. The refined conceptual framework is shown in figure 2.

Based on the refined conceptual framework, each eligible study was coded as being associated with one or more of the three phenomena of interest, namely slum residents' healthcare accessing (which covered perception of needs/desire for care, healthcare seeking and healthcare reaching as defined by Levesque *et al*[34]), healthcare utilisation and provision of health services (which covered various arrangements related to service delivery) in slum settings (figure 2).

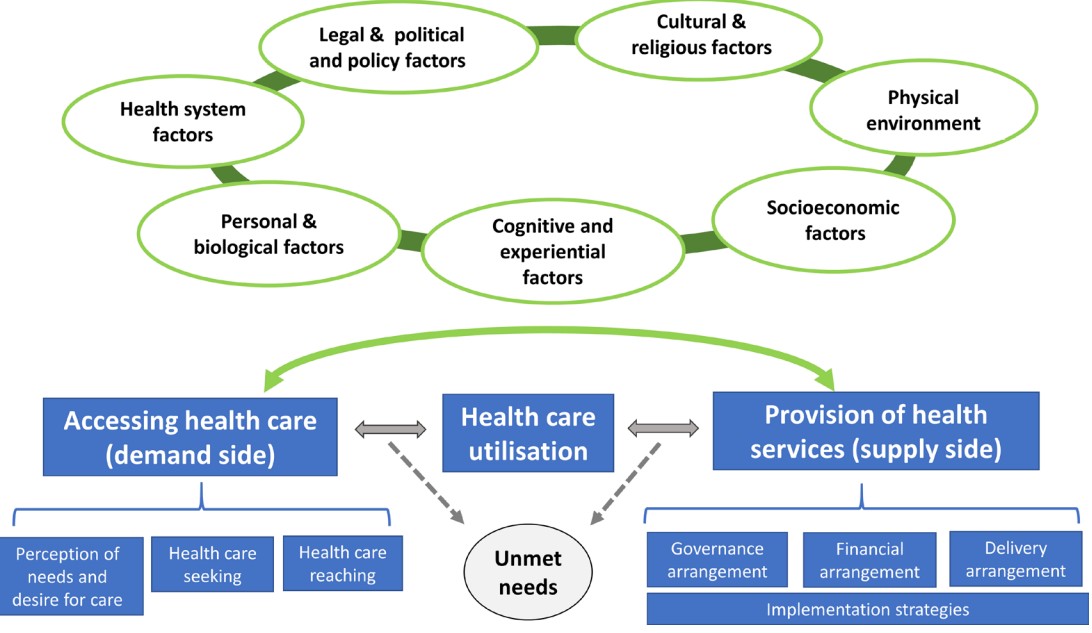

**Figure 2** Updated framework of factors influencing healthcare-seeking behaviour/healthcare utilisation/provision of healthcare services in slums.

In addition, to facilitate the organisation of complex evidence in this review, diverse factors reported in the included studies were initially classified into six different categories according to the preliminary framework shown in figure 1. However, during the data-charting process, we realised that many factors such as perception of symptoms and experience from past use of healthcare services did not fit into one of these six original categories. A new category of 'cognitive and experiential factors' was therefore added to the refined conceptual framework (figure 2) to reflect the emerging themes, which include seven categories:

1. Personal and biological factors: these relate to personal characteristics of slum residents, including age, sex, ethnicity and the nature and severity of health conditions.
2. Cognitive and experiential factors: these relate to personal awareness, knowledge, perception, attitude, belief and experience, etc, formed through cognitive process based on upbringing and past events.
3. Socioeconomic factors: these include income and wealth, economic hardship/poverty and economic opportunities, marital status, education, crime, social capital (such as bonding, trust and reciprocity between close relatives, neighbours and community members),[36] use of technologies for social and economic purposes, commercial and charitable organisations and activities.
4. Physical environment: this covers natural environment such as proximity to a health facility, built environment and infrastructure such as water supply, transport and mobile/internet networks, as well as weather conditions and environmental pollutions.
5. Cultural and religious factors: these include cultural and religious beliefs and activities, and local and national customs.
6. Legal, political and policy factors: these include government policies and issues related to legal, justice and political systems.
7. Health system factors: these relate to historical and current organisation and provision of healthcare that may impact on provision and delivery of health services in individual slum communities and the services experienced by slum residents.

In addition to the 'cognitive and experiential factors' category, another major difference between the preliminary (figure 1) and refined (figure 2) conceptual framework relates to the definition of healthcare access. Our preliminary framework adopted the definition by Levesque and colleagues, who defined healthcare access as 'the possibility to identify healthcare needs, to seek healthcare services, to reach the healthcare resources, to obtain or use health care services, and to actually be offered services appropriate to the needs for care'.[34] However, during our study-screening and data-charting process, we found that it would be helpful to make a distinction between the process of 'accessing' healthcare (which covers gaining awareness of needs, forming

an intention to seek healthcare and taking an action to reach healthcare) and the actual receipt and utilisation of healthcare ('accessed care') when examining empirical evidence, as healthcare needs could only be met when the latter occurs and this depends on both factors related to service users (demand side) and factors related to service providers/planners (supply side). Therefore, we separated out utilisation of healthcare from 'accessing health care' to highlight that it requires a match between demand and supply side factors.

Data on study population, study design, country in which the study was conducted, methodology and associated factors were extracted using a data-charting spreadsheet which was developed and continuously updated as the review progressed by two of the reviewers (J-EP and Y-FC). Whether a study was conducted exclusively within slums and whether a comparison was made between slum and non-slum urban or rural residents were also noted. Coding of phenomena and factors and data charting were conducted by one reviewer (J-EP) and checked by a second reviewer (PK, GY, OO, Y-FC). Disagreements were discussed between reviewers until consensus was reached.

### Patient and public involvement
Given the focus of this scoping review on published literature, we did not directly involve residents and service providers/planners from slum settings. Nevertheless, our wider project has a work package that specifically engages with slum residents and service providers and planners,[28] and early plans and findings of this review were shared with the wider project team who provided comments based on their experiences of community engagement.

### RESULTS
The reporting of this review follows the Preferred Reporting Items for Systematic Reviews and Meta-Analyses extension for Scoping Reviews.[37] Using the search strategy described earlier, a total of 15 469 records were retrieved from the initial and updated searches (MEDLINE 4688, Embase 5090, Web of Science 3553, Cochrane 381, CINAHL 1757), with 9916 records remaining after excluding duplicates. Two additional articles[18 38] were identified from references of the included studies. As described earlier, screening was limited to the 4368 records published from 2016 onwards.

A total of 111 articles were included in this scoping review (figure 3). Thirty-two studies reported factors associated with healthcare accessing of slum residents, 73 studies reported factors related to healthcare service utilisation and 10 articles reported the factors related to provision of healthcare services in slums (four studies reported factors related to more than one phenomenon of interest). Seventy-four of the 111 studies were quantitative studies, 21 studies were qualitative studies and 14 studies were undertaken using mixed methods. The remaining two studies were systematic reviews. A total of

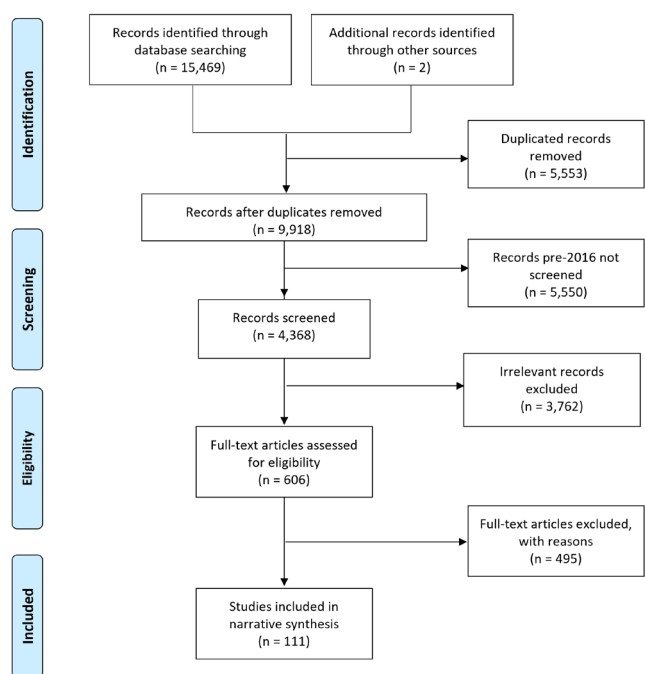

**Figure 3** Flow chart.

42 (38%) studies were conducted in India, followed by Kenya (14 studies, 13%) (table 1).

Participants, country, study design, methodology, observed phenomena and outcomes, and factors of interests for each study are described in online supplemental tables 1–3. Online supplemental table 1 shows 32 studies reporting factors associated with general healthcare-seeking behaviours; healthcare seeking for children or women; slum residents' preference for healthcare providers; and healthcare seeking related to HIV testing. Online supplemental table 2 presents various factors reported in 73 studies related to general healthcare utilisation as well as use of specific services such as childhood immunisation, maternal healthcare and possession of health insurance. In online supplemental table 3, ten studies reporting factors related to the provision of health services in slums are summarised. Key findings are described below.

### Demand side: factors associated with healthcare accessing and healthcare utilisation of slum residents

We found 104 articles which identified many different factors affecting healthcare accessing and utilisation. These factors are often inter-related and exert their influence at different levels (eg, from personal, family to community level) in different circumstances. We classified various factors into seven categories (figure 2). Factors particularly relevant to slum settings and other commonly identified factors within each category are highlighted below.

Personal and biological factors: The common factors associated with healthcare accessing and utilisation included intrinsic factors such as age,[21 39–55] sex[18 21 41 45 51 53 55–58]

**Table 1** Characteristics of included studies

| Category | Subcategory | | Studies, n (%) |
|---|---|---|---|
| Publication year | 2016 | | 22 (20) |
| | 2017 | | 17 (15) |
| | 2018 | | 23 (21) |
| | 2019 | | 22 (20) |
| | 2020 | | 18 (16) |
| | 2021 | | 9 (8) |
| Analysis method | Quantitative | | 74 (67) |
| | Qualitative | | 21 (19) |
| | Mixed methods | | 14 (13) |
| | Narrative synthesis | | 2 (2) |
| Study location | Asia | India | 42 (38) |
| | | Bangladesh | 9 (8) |
| | | Nepal | 4 (4) |
| | | Pakistan | 3 (3) |
| | | Myanmar | 2 (2) |
| | | Iran | 2 (2) |
| | | Sri Lanka | 1 (1) |
| | South America | Brazil | 7 (6) |
| | | Peru | 2 (2) |
| | Africa | Kenya | 14 (13) |
| | | Ethiopia | 7 (6) |
| | | Malawi | 4 (4) |
| | | Uganda | 3 (3) |
| | | South Africa | 2 (2) |
| | | Sierra Leone | 1 (1) |
| | | Nigeria | 1 (1) |
| | | Egypt | 1 (1) |
| | | Zambia | 1 (1) |
| | | Namibia | 1 (1) |
| | | Ghana | 1 (1) |
| | North America | Haiti | 1 (1) |
| | Multiple nations | | 1 (1) |
| Healthcare services in slums* | Healthcare accessing | | 32 |
| | Healthcare service utilisation | | 73 |
| | Provision of healthcare services | | 10 |
| Total | | | 111 (100) |

*One study reported factors related to both healthcare accessing and healthcare utilisation and three studies reported factors related to both healthcare utilisation and provision of healthcare services.

and ethnicity,[21] familial factors such as birth order of the sick child,[21 46 59–61] as well as personal health and type of illness,[45 58] disability[47] and morbidity,[21 51 62 63] and the specific features of the health condition.[52 55 64] Slum residents are more likely to seek healthcare services when sick

children are younger,[48 49 52 55] but evidence on the association between mother's age and child's vaccination was inconsistent.[21 39] Healthcare seeking and utilisation were different by sex, but the association was context dependent. Several studies reported higher healthcare utilisation among female slum dwellers,[18 51 55 57] while other studies showed male children had higher vaccination coverage[56] and incurred more medical expenditure.[58] Major life events such as recent migration[21 49 65–67] and relocation[48] into slums tend to be associated with lower healthcare seeking and utilisation. Recency of migration to slums was also related to lower uptake of Rashtriya Swasthya Bima Yojana (RSBY), a national health insurance programme run by the Indian government for poor families.[68] People with specific symptoms (such as fever, tachypnoea, persistent vomiting),[52 55 64] disability[47] and illnesses including chronic disease[21 51 53 62 63] tend to use healthcare services more. Although people with tobacco habit were less likely to participate in breast cancer screening, they were more likely to take part when they had family history of cancer or history of cancer screening.[69] Lower birth order of the child was associated with increased utilisation of hospitals for childbirth,[21 46 59–61] while the use of family planning service[46] and out-of-pocket expenditure was higher in multigravida than primigravida.[62]

Cognitive and experiential factors: These factors were not included in our initial conceptual framework but rather emerged inductively from our data. Consequently, their identification led us to revise the conceptual framework for this scoping review. A wide range of factors formed through cognitive processes and influenced by individual's upbringing, past experience and surrounding environment were reported to be associated with both healthcare seeking and healthcare utilisation of slum residents. Perception,[39 50 52 67 70–76] knowledge[52 77–82] and experience of symptoms and illnesses[50] were commonly found to influence healthcare seeking and utilisation. Mothers who experienced child death and subsequently planned pregnancy showed higher use of antenatal healthcare services.[50] When people perceived the symptom or disease to be serious they tend to seek healthcare services.[52 70 72 76] Although lack of knowledge could be a barrier to accessing healthcare services,[77 78] one study showed caregivers with good knowledge of child danger signs were less likely to seek healthcare services timely.[52] People perceiving their health status as good showed lower odds of having insurance,[83] but awareness and knowledge of health problems led people to use healthcare services.[39 73 75 79 80 82] Home remedy and home management delayed healthcare-seeking behaviour.[52 70 81 84] In addition, perception,[21 42 63 75 83 85–89] knowledge[21 42 48 60 67 74 81 85 90–95] and experience of healthcare services[39 49 59 61 66 69 74 79 86 96 97] including fear and distrust of healthcare services,[21 38 67 71 74 75 78 88 98–100] and preference related to care provider's gender[87 101] were frequently cited factors. Provider shopping associated with distrust of healthcare providers and denial of diagnosis delayed first care seeking and treatment initiation of

patients with pulmonary tuberculosis in India.[70] Perception or experience of healthcare services also affected uptake or renewal of health insurance.[83 94]

Socioeconomic factors: Socioeconomic status was associated with utilisation of healthcare services,[21 39 40 46 56 61 81 102] and even though one study showed that slum residents of lower socioeconomic class were more likely to enrol in health insurance than slum residents of higher socioeconomic class,[40] the latter were more likely to use healthcare services.[39 46 56 61] One study reported higher public hospital visits (compared with private hospital visits) among lower socioeconomic status.[103] Income and wealth[21 41 48–50 60 65 67 80 104–108] including financial constraint[19 21 38 45 73 76 78 84 86 89 100 102 109–113] featured prominently. Higher education level[39 48 55–58 60 61 65–67 69 80–82 106–108 114–116] and higher income[21 48–50 55 57 60 65 67 80 104 106–108 117] were associated with more seeking and utilisation of healthcare services. With some exceptions,[66 104] previous studies reported that employed slum residents tend to seek and use healthcare services more frequently than unemployed slum residents and housewives.[48 50 65 66 69 80 82 83 118] Even though married people tend to seek and use more healthcare services,[18 69] the reported influence of family type was inconsistent.[39 50 115] Female slum residents in nuclear family used more antenatal services than those in joint family type,[50] but female slum residents in joint family type used more postnatal service[39] and immunisation service for their children.[115] Smaller family size used more maternal healthcare services,[66] and bigger households had higher odds of having health insurance.[68] The socioeconomic challenges faced by slum residents also manifested as competing priorities[73 93 119] and lack of time[21 100 120] for healthcare seeking and utilisation, because they did not want to or could not afford to miss work and lose income,[21 70] which can be exacerbated by lack of social support.[74 76 87 93 99 121]

Physical environment. Slum residents considered proximity of healthcare facilities,[21 38 46 76 80 84 85 95 108 122–126] transport such as travel assistance,[76] lack of transportation,[38 82 102] traffic congestion[127] and environment of residence area when they sought and used healthcare services. Long distance from health facility,[38 54 76 84 101 122] no transportation or travel assistance[38 82 102] and unsafe environment of residential area such as darkness at night were reported as barriers to reaching healthcare facilities.[99]

Cultural and religious factors: These included religion[41 56 59 69 75 114 128]; sociocultural influence[93 101] such as exposure to media[79 97]; stigma associated with unplanned/extramarital pregnancy,[78 93] postpartum depression[129] and other illnesses such as contagious skin disease, barrenness and female sexually related problems[101]; and use of traditional/home medicine.[75 86 99] Women in slums could not go to hospital because they had difficulties in disclosing the symptoms, postponed their health issues because of their responsibilities at home and engaged in self-treatment practices such as home remedies recommended by grandmother and friends because

of sociocultural influences towards healthcare-seeking behaviour.[77] Women in Ethiopia reported not returning to postnatal care due to religious and cultural expectation for the mother and the baby to stay home for 80 days after birth.[93] One Indian survey showed that some women could not seek healthcare services during labour since their husband or family did not allow that.[38]

Legal, political and policy factors: Type of slums (in terms of official recognition and availability of basic facilities) and possession of a ration card were found to be associated with uptake of the Indian RSBY national health insurance programme.[68] One study reported that slum residents could not seek healthcare facilities for abortion because of the perceived illegality of abortion.[78]

Health system factors: Slum residents were also influenced by many factors related to health systems when they sought healthcare. These included accessibility associated with the location[21 101] and timing of services[21 84 86]; quality of healthcare services[38 66 85 86 101 112 124] such as delay in advising patients to go for related tests or referral,[70] likelihood of receiving appropriate examination[84 109] and adverse events.[75] Slum residents considered service organisation including medical turnover,[130] availability of supplies/healthcare workers,[46 84 109 112] attitude of healthcare providers,[86] type of healthcare facilities[38 85 116 131] and waiting time.[72 84-86 109 111 132] Slum residents tend to seek government and non-governmental organisation facility[117] and avoid private hospitals[38] for healthcare services. Healthcare utilisation was higher among slum residents with healthcare insurance than those without it,[18 82] and households with higher quarterly out-of-pocket healthcare expenditure had lower scores for an index of access to primary healthcare.[133]

In an Ethiopian study, some participants reported unavailability of female birth attendants as a reason for not delivering at healthcare facilities[112] (table 2).

## Supply side: provision of healthcare services

Ten articles described factors associated with provision of healthcare services in slums from the service providers' perspective. None of the studies reported personal and biological factors. Factors related to other categories are summarised below.

Cognitive and experiential factors: Odhiambo *et al* reported slum residents' fear of side effects, size of tablet and misconceptions regarding treatment as the factors hindering drug administration activities by healthcare workers for a deworming programme in Kenya.[134] On the other hand, this study also reported a high demand for drugs from slum residents in the final year of this programme because people realised that free treatment was to be ended.[134]

Socioeconomic factors: Effective community mobilisation was a facilitator[134] whereas poor community support[135] and insufficient time allocated for providers to implement healthcare programmes[134] were barriers for provision of healthcare services in slums. In the deworming programme mentioned above, community

health workers reported that direct observation of slum residents taking deworming drugs after meals was sometimes not feasible because slum residents skipped meals or ate late at night due to food shortage.[134] Some slum residents demanded money to take the deworming drugs, either to facilitate purchase of food or to have their own share of the money that they perceived the community health workers would be paid by the programme if they complied with taking the drugs.[134]

Physical environment: Poor sanitation,[134 136] presence of rodents and no pavement[136] and bushy and unprotected environment[134] were reported as factors making the provision of healthcare services difficult in slums.

Cultural and religious factors: Religious beliefs and mistrust of interventions,[134] lack of a shared understanding of the needs, purposes and consequences of family planning and pregnancy-related services among slum residents and healthcare providers[73] were the barriers for healthcare service provision. In the previous deworming programme, portrayal of unrelated death being linked to the programme and related negative publicity affected participants' compliance.[134]

Legal and political factors and policy: Devolution of service delivery through downward transfer of funds and responsibilities from central/national government level to elected local bodies; management by professional managerial and technical cadres; tight organisation of public health services; and professional support from the state directorate of public health were found to strengthen public health service provision in Chennai slums compared with Delhi.[137] One study reported that policies affected healthcare provision negatively because of staff shortage arising from change and suspension of the appointment of health promoters, which led to overwork and lack of time to provide required care by other healthcare staff.[132] In Brazil, home visits for the provision of healthcare services were hampered because slum residents could not present documents required to register for healthcare.[136] On the other hand, giving priority to socially less developed areas for strengthening the Family Health System in Brazil might have been associated with better service coverage for slum residents with tuberculosis compared with their urban non-slum counterparts.[138]

Health system factors: Pay scale of frontline healthcare workers,[135] knowledge of intervention area by community health workers,[134] issues related to rigid task assignment by service managers,[136] requirement to follow standardised protocol,[136] demands from the management,[136] work burden[132 136] and no incentive,[135] insufficient time,[134] attitude[73] and support of healthcare providers,[135] ill-defined geographical boundary of service with unserved areas and left-out urban slum pockets[139] were associated with healthcare service provision in slums.

Lack of community-based care (such as school-based education for reproductive health and community support networks for women),[132] unreliable immunisation and household data[128]; and inefficient utilisation of funds,[128] affordability (price) and availability of

**Table 2** Factors associated with healthcare accessing and healthcare utilisation in slums from service user's (demand side) perspective

| Factors | Healthcare accessing | Healthcare utilisation |
|---|---|---|
| **Personal and biological factors** | | |
| Age | (−) Age[52–55] | (±) Age[21 39–51]; (+) age of household head[18] |
| Gender | (±) Sex[53 55 57 58] | (±) Sex[18 21 41 45 51 56 58]; (male) sex of household head[133] |
| Ethnicity | | Ethnicity[21] |
| Migration | | (−) Recent migration[21 49 65–68]; (−) relocation[48]; (−) return to home village[21] |
| Biological | (+) Symptoms such as fever, tachypnoea, chest in drawing, persistent vomiting[52 55 64]; having disease[53] | Type of illness[45 58]; (+) having a disability[47]; (+) morbidity[21 51 62 63] |
| Other personal | (−) Tobacco habits[69]; (+) family history of cancer and history of cancer screening[69] | (−) Birth order of sick child[21 46 59–61]; (−) parity[42 46 62 148] |
| **Cognitive and experiential factors** | | |
| Knowledge/experience of symptoms and illnesses | (+) Perception of symptoms[70] or illness[52 72 76]; (±) knowledge of symptom/disease[52 77 78]; (−) denial and complacency[71] | (+) Experience of child death[50]; (+) planned pregnancy[50]; (+) perceived health status[83] and health problem[39 73 75]; (+) knowledge of symptom[82]; disease[79 80] |
| Ability/experience in handling health-related conditions and perceived needs for accessing health services | (+) Awareness of the need for healthcare services[38 53 55]; (−) home remedies[70] or management of childhood illness[52 84] | (+) Perceived needs for healthcare services[21 67 74 75 90 93 112 128]; (−) home delivery[81] |
| Perception/knowledge/experience/preference of health services | (−) Fear of mistreatment[71 100] and (−) doubts about medical care[38 78 99]; gender-induced affordability[101]; (−) provider shopping[70] | (positive) Perception of healthcare services[21 42 63 75 83 85–89] and providers[21 84 93 111 124]; (+) knowledge of health services[21 42 48 60 67 74 81 85 90–93] or facilities[21 94 95]; (+) previous use of related healthcare services[39 42 49 59 61 66 79 96 97]; (−) bad experiences of friends and relatives at healthcare facilities[94]; (−) misunderstanding or fear[21 67 74 75 88 98 128]; gender healthcare worker preference[87], (−) side effect[81]; lack of trust[45] |
| **Socioeconomic factors** | | |
| Socioeconomic status | (−) Social class[103]; social group (caste) of caregiver[53] | (+) Socioeconomic status[21 39 40 46 56 61 81 102]; caste[108 114]; (−) insecure or poor residential background[21 46 68 81]; (+) possession of ration card[68] |
| Marital status | (married) Marital status[69] | (married) Marital status[18 41]; duration of marriage[42] |
| Family composition and living arrangement | (−) Family size[53] | (±) Family type[39 50 115 126]; (±) family size[66 68]; (−) number of children in household[21 44 48]; (+) number of male children[149]; (+) housing condition[21] |
| Education | (+) Education[53 55 57 69] | (+) Education[39 41–43 48 56 58 60 61 65–67 80–82 106–108 114–116 126 148]; (±) husband education[44 50]; (+) mother's education and literacy[21 43 46 56 59] |

**Table 2** Continued

| Factors | Healthcare accessing | Healthcare utilisation |
|---|---|---|
| Income and wealth | (+) Income[55 117]; (+) wealth[54 57]; (−) inability to afford care[19 38 76 78 84 89 100 109 110] | (+) Income[41 49 50 67 80 104]; (+) wealth[21 48 60 65 105–108]; (−) financial constraint[21 45 73 86 102 111–113] |
| Occupation | (+) Occupation[53 57 69] | (+) Employment[21 65 83 118 126]; (±) occupation[48 66 80 82 104 148]; (±) occupation of spouse[50 60] or household[68] |
| Social support | (−) Difficulty in reaching services (security risk at night)[99]; (+) accompanying person[76]; decision-making person for seeking healthcare[53] | (+) Family support[74 87]; (+) social connectedness[93]; (+) socioeconomic support[121]; permission for immunisation by decision-maker[128] |
| Competing priorities/lack of time | (−) Competing priorities (ability to work and income)[119]; (−) not want to miss work[70]; (−) lack of time[100 120] | (−) Competing priorities[73 81 93]; (−) risk of lost income[21]; (−) parents being too busy[21] |
| Physical environment | | |
| Distance from health facility | Proximity of healthcare facilities[38 54 76 84 122]; geographical distance of formal healthcare[101] | (−) Distance from health facility[21 46 80 85 95 108 123–126] |
| Transport | (+) Travel assistance[76]; (−) no transportation[38] | (−) Lack of transportation[82 95 102]; (−) variability in traffic congestion[127] |
| Environment of residence area | (−) Difficulty in reaching services (darkness at night)[99] | Residential background[21 68 105] |
| Cultural and religious factors | | |
| Religion | Religion[69] | Religion[41 56 59 75 114 128] |
| Sociocultural influence | (−) Stigma[78 101 129]; mother tongue[69]; (−) difficulties in disclosing the symptoms, (−) neglecting behaviours and sociocultural influences[77]; (+) cultural competency of care[101]; (+) easy communication[101]; living with the burden of cultural expectations[101]; (−) no permission to seek care from family[38] | (−) Exposure to media[79 97]; stigma[128]; (−) cultural expectation for women after birth and fear of stigma for pregnancy out of wedlock[93] |
| Tradition | (−) Traditional medicine[99] | (−) Traditional remedies[75]; (−) home remedies[86] |
| Legal, political and policy factors | | |
| Legal issues | (−) Perceived illegality of abortion[78] | Type of slums and possession of a ration card[79] |
| Health system factors | | |
| Accessibility | (+) Ease of access[101]; (−) late facility opening times[84] | (−) Limited access to the services due to location[90 93]; (−) timing of services[21 86]; household visit by health workers[21] |
| Quality and safety of services | Quality of treatment and expected outcome of therapies[38 101]; (−) delay in advising related tests[70]; referral[70]; optimal examination[84 109]; (−) provider shopping[70] | Quality of service[66 85 86 91 112 124]; (−) adverse events[75] |
| Charges for health services | (+) Insurance coverage of both public and private providers and of extended family members[150] | (−) Average out-of-pocket healthcare expenditure[133]; healthcare insurance[18 63 82] |

Continued

**Table 2** Continued

| Factors | Healthcare accessing | Healthcare utilisation |
|---|---|---|
| Service organisation and delivery arrangement | (–) Medical turnover and overload or healthcare providers[130]; (+) government/NGO facility[117]; (–) private hospital[38]; early engagement by healthcare workers[54] | Attitude of healthcare providers[86] [95]; mode of delivery[39 56 59 62 81 151]; (–) hospitals refused to accept health insurance cards[94] |
| Facility and resources | Availability of medicines and supplies[84 109]; (–) lack of healthcare facilities[144] | Type of healthcare facility[40 85 95 116 131] [133]; inadequate resources[90]; (+) number of available healthcare workers[46]; (–) unavailability of female birth attendants[112] |
| Waiting time | (–) Waiting time[72 84 109] | (–) Waiting time[85 86 111 132] |

(–) Negative association; (±) inconsistent/conflicting evidence or context dependent; (+) positive association.
NGO, non-governmental organisation.

medicine,[140] limited medical supplies[73 135] and infrastructural facilities,[135] inadequate space and equipment,[136 139] suboptimal training of staff,[139] insufficient availability of logistics and health manpower[139] also affected service provision (table 3).

### Comparison between slums and other settings

Seven studies which met our inclusion criteria also included data from non-slum urban and/or rural areas and potentially allowed exploration of factors associated with healthcare access across different settings. Key findings from these studies are summarised in table 4.

These recent studies showed a mixed and dynamic picture of healthcare access across slum and other settings and reported various factors associated with this. For example, the proportion of young children fully immunised was found to be lower in slums compared with non-slum urban setting but was higher than rural settings in Nigeria. Nevertheless, the coverage improved over time across all settings.[59] While many common factors associated with full immunisation of young children were identified, giving birth in health facilities (as opposed to home) had a larger positive effect on subsequent immunisation coverage in slums compared with non-slum urban and rural settings.[59] A narrowing of gaps in delivery by skilled birth attendants between slum and non-slum urban settings over time and a reverse of the trend from having lower usage to higher usage of modern contraceptive methods by married women in slums versus urban non-slums were reported in Bangladesh.[46] Slum residents reported financial issues being the main reason for not taking prescribed drugs whereas getting better was the cited main reason for urban non-slum residents in Iran.[113] Better coverage of services and higher rates of treatment completion were reported for patients with tuberculosis in slums compared with non-slum urban setting in two studies in Brazil,[51 138] where a higher priority given to enhancing the Family Health System in socially less developed areas in recent years was suggested to be

a likely factor associated with better service provision in slums[138] (table 4).

### DISCUSSION

#### Statement of principle findings

This scoping review of recent literature examined the demand side factors associated with slum residents' healthcare accessing and utilisation, as well as supply side factors associated with provision of health services in slums. We found over 104 studies related to the former but only 10 studies related to the latter. We identified different factors associated with accessing, utilisation and provision of health services in slums, and mapped them to a conceptual framework developed and refined for this review into seven broad categories (figure 2).

#### Findings in the context of existing literature

Even though previous reviews have investigated factors associated with healthcare access in various settings,[141 142] to our best knowledge this scoping review is the first that has examined wide-ranging factors across different service areas of healthcare in slums. Our findings are consistent with previous studies which highlighted common factors associated with healthcare seeking and utilisation such as age, income and education.[141 143] We identified several factors that are particularly pertinent in slum settings, such as costs of healthcare,[19 21 73 76 78 84 89 100 102 109–111] lack of time due to slum residents' competing priorities[21 100 120] and issues arising from adverse physical environment,[82 102 134 136] security,[99 136] fear of formal registration due to distrust of the authorities[136] and proximity of healthcare facilities.[21 76 80 84 85 108 122–125] In addition, included studies showed that the effects of a given factor may differ between slum, urban non-slum and rural settings.[59]

Healthcare cost is a major barrier between the intention to seek care and actual utilisation of services.[109 144] Healthcare provision supported by tax-based financing and/or various forms of social and private insurance

**Table 3** Factors associated with provision of healthcare services in slums from service provider's (supply side) perspective

| | |
|---|---|
| **Cognitive and experiential factors** | |
| Perception/knowledge/experience/preference of health services | Fear of side effects, size of tablet and misconceptions regarding treatment, high demand for drugs in the final year of treatment[134] |
| **Socioeconomic factors** | |
| Income and wealth | Difficulty in directly observing deworming treatment at mealtime due to food shortage[134] |
| Social support | Effective community mobilisation[134]; poor community support[135]; non-involvement of community members and urban local bodies[139]; absence of community members during the drug administration exercise[134]; demand for incentives by community members to take deworming drugs[134] |
| **Physical environment** | |
| Environment of residence area | Environment (sanitation, territory)[136]; unsanitary environmental conditions[134]; inaccessibility (filthy and bush environment)[134] |
| **Cultural and religious factors** | |
| Religion | Religious beliefs and mistrust of interventions[134] |
| Sociocultural influence | Lack of shared understanding of the problems in community[73]; unrelated death and the associated negative publicity (of a deworming programme) by the media[134] |
| **Legal, political and policy factors** | |
| Policy issues | Devolution of service delivery transferring funds and responsibilities to elected local bodies[137]; management by professional managerial and technical cadres[137]; tight organisation of public health services[137]; professional support from the state directorate of public health[137]; healthcare policies[132]; policy prioritising low social development areas[138] |
| Legal issues | Fear of requirement for formal registration[136] |
| **Health system factors** | |
| Cost | Pay scale of frontline healthcare workers[135]; medicine price[140] |
| Quality and safety of services | Knowledge of intervention area by community health workers[134] |
| Service organisation and delivery arrangement | Issues related to assignment of tasks[136]; requirement to follow standardised protocol[136]; demands from the management[136]; work overload[132 136]; underperformance of staff[128]; documentation work/work burden/no incentive for work[135]; insufficient time[134]; attitude of healthcare providers[73]; lack of supportive staff[135]; community health worker familiarity with households led to warm reception[134]; opportunity to integrate mass drug administration with other health interventions[134]; presence of community health workers and their supervisory structure, and points of referral for serious side effects[134]; restriction of range of services[139]; unserved areas and left-out urban slum pockets[139]; poor monitoring and supervision[139]; unreliable immunisation and household data[128] |
| Facility and resources | Community-based care[132]; inefficient utilisation of funds[128]; affordability and availability of medicine[140]; limited medical supplies[73 135]; infrastructural facilities[135]; inadequate space and equipment[136]; suboptimal training of staff[139]; insufficient availability of space, logistics and health manpower[139] |

that reduce out-of-pocket expenditure at point of care could be potential measures to overcome this barrier and help achieve universal coverage goals. Limited evidence showed that initiatives prioritising primary healthcare coverage in slums could improve access,[138] but there is insufficient evidence from studies included in this review to determine the best model of healthcare financing for improving healthcare access and coverage in slum settings.

Although possession of/coverage by health insurance was associated with higher levels of utilisation of health services among slum residents,[18 82] studies showed that uptake of government-run public insurance among slum residents was low.[68 83] This may be attributed to lack of awareness, difficulties in navigating through the health system and in obtaining official proof of identity required for enrolment[68] and poor quality of care and range of services offered.[68 83] Even among slum residents covered

**Table 4** Studies that examined factors associated with healthcare seeking and utilisation in both urban slum and non-slum urban and rural settings

| Study and location | Differences in healthcare access | Associated factors |
|---|---|---|
| Kalyango et al[150] Kampala City, Uganda | *Preferences and willingness to pay for health insurance* Households in non-slum communities had a high preference for health insurance plans covering chronic illnesses and major surgeries to other plans. | Coverage of extended family (vs restricted enrolment of children); coverage of both private and public providers (vs private only). |
| Obanewa and Newell[59] Nationwide, Nigeria | *Fully immunised child coverage (FIC)* Proportion in slum lower than urban non-slum but higher than rural; proportions increased between 2003 and 2013 across all three settings. | From multivariable regression*: year, birth order, antenatal attendance, maternal education level, religion, maternal age at child's birth, media exposure, region of the country, interaction between place of residence and place of delivery. |
| Angeles et al[46] Multiple cities, Bangladesh | *Use of modern contraceptive methods* Proportion changed from being lower in slums in 2006 to being higher in slums in 2013 compared with urban non-slums. | From multivariable regression*: parity, mother's age, mother's educational attainment, socioeconomic status, interaction (slum×time period). |
| | *Delivery by skilled birth attendant* Proportion substantially lower in slums compared with urban non-slums but the gaps narrowed over time. | From multivariable regression*: residing in slums, parity, mother's age, mother's educational attainment, length of stay in current city of residence, socioeconomic status, number of available community health workers, distance from health facility, interaction (slum×time period). |
| Islam[106] Multiple cities, Bangladesh | *Antenatal care visits* 'there was a large inequality' between slum and urban non-slum (detail not reported). | Level of educational attainment, wealth index of the household. |
| | *Using contraceptive methods* 'Prevalence rate higher among slum women' than urban non-slum women. | Not reported. |
| Tabrizi et al[113] Tabriz, Iran | *Utilisation of health services in the past 30 days* Similar utilisation overall, but with lower proportion received needed health services and used private clinics, higher use of vaccination and maternal health services, and lower use of services for heart failure and hypertension for slum residents compared with urban non-slum. | High cost of services. |
| | *Home care services* Very little use both in slum and urban non-slum areas. | High cost of services. |
| | *Prescribed drug during last visit to health facilities* Lower proportion for slum versus urban non-slum. | Not reported. |
| | *Not taking drugs prescribed* Higher proportion for slum versus urban non-slum. | Main reason: financial problems for slum versus getting better/feeling well for non-slum urban. |
| Snyder et al[51] Rio de Janeiro, Brazil | *Directly observed treatment coverage for tuberculosis (TB)* Higher for slum versus urban non-slum patients with TB. | Not examined. |
| | *Abandonment of TB treatment* Lower for slum versus urban non-slum patients with TB. | From multivariable regression*: residency in a slum, sex, age, extrapulmonary clinical disease, HIV/AIDS, interaction (directly observed treatment×residency in a slum). |
| Prado Junior et al[138] Rio de Janeiro, Brazil | *Coverage under Family Health System for patients with TB* Higher for slum versus urban non-slum. | Giving the Family Health Strategy priority to coverage of areas with lower social development. |

*From the model with most comprehensive adjustment including residency in slum as one of the variables; only factors that were statistically significant (at 5% level) are shown.

by health insurance, access to care was often refused and additional charges were frequently requested.[94] Policies that aim to improve access to healthcare services among slum residents through public health insurance will need to address these challenges.

Several studies reported lack of time and competing priorities as a factor affecting healthcare-seeking behaviour[100 119 120] and health service utilisation.[21 73 93]

This suggests a delicate balance between factors that individual slum residents have to strike when making decisions on healthcare seeking and utilisation. van der Heijden et al showed that health was considered as an asset for working ability in slums,[119] but paradoxically the ability to work often seems to impede healthcare seeking for health issues. This highlights the importance of considering slum residents' interest and priorities when

providing healthcare services and promoting healthcare utilisation in slums.

## Strengths and weaknesses of the review

This scoping review has several strengths. We conducted a comprehensive literature search using generic terms related to slums with few other restrictions. The search was therefore likely to be sensitive for identifying relevant literature. Contemporary methodological guidelines for undertaking scoping reviews were followed,[22] and a conceptual framework which was adapted based on emerging findings was used to facilitate the organisation of evidence.

The review has enabled theory building and refinement of a conceptual framework. Our preliminary framework included six categories (figure 1). During data coding and extraction, it emerged that many studies reported perception, knowledge and experience of slum residents being associated with their healthcare seeking and utilisation. We subsequently classified these factors as cognitive and experiential factors, which primarily consist of three subcategories: knowledge/experience of illness, perceived needs for accessing healthcare services and perception/experience of healthcare services. These factors were influenced by other factors included in our original conceptual framework, but highlighted the crucial links between those factors and the ultimate actions by individual slum residents to access health services. Future interventions to promote health service utilisation for slum residents[145] could make use of our framework to develop programme theories and map out causal pathways.

This review also has some limitations. Given time constraint, we were only able to examine the most recent literature published in English in academic journals, and have not examined the methodological quality of individual studies (which we noted to be quite varied) in detail. We attempted some preliminary synthesis to configure the identified evidence but have not explored the complex relationship between the factors identified and their interplay with the context of individual slums in depth. Nevertheless, the findings from this scoping review will provide a good foundation for further syntheses.

## Methodological considerations

A number of challenges in the process of classifying and coding data are worth mentioning. First, access to healthcare has been conceptualised and defined in various ways in previous studies. The WHO suggested six building blocks of a health system including service delivery, health workforce, health information systems, access to essential medicines, financing and leadership/governance to strengthen health systems[146] and, in its report, defined access to healthcare as public responsibility for ensuring all citizens' entitlements to the protection of their health beyond simply a proportion of a target population that benefits from an intervention, towards universal coverage.[146] They also pointed

out system constraints such as financial access difficulty, physical access difficulty, low knowledge and skills, poorly motivated staff, weak leadership and management, ineffective intersectoral action and partnership as barriers to access.[146] The WHO's definition and conceptual framework focus on health system-level factors and would be particularly useful when examining supply side factors, which seem to be understudied based on our findings. As described in the Methods section, we primarily adopted the conceptual model of healthcare access developed by Levesque and colleagues given our shared focus on service users. However, in our conceptual model, we separated the dynamic stages of 'accessing' healthcare from the actual 'accessed' healthcare (utilisation) to highlight the crucial match required between the demand side and supply side factors to facilitate utilisation of healthcare when there is a need.

Several factors associated with healthcare accessing and utilisation can be viewed from different perspectives and therefore potentially be coded under different categories. For example, barriers for healthcare seeking and utilisation related to costs can be considered as socioeconomic issues from the slum dwellers' perspective but can also be viewed as health system issues for not offering the services in an affordable way. Indeed, previous access frameworks suggested that access is created and negotiated in a dynamic interchange between households/communities and healthcare workers/systems (ie, demand and supply) on each access dimension.[34 147] In such situations, we tried to code a factor under the category that most directly reflects the original data through discussions within the review team (in the example of healthcare cost, the factor was coded primarily under socioeconomic factors rather than health system factors when the factor was reported by slum residents as a barrier); otherwise, more than one category was coded (eg, bad experience from previous utilisation of health services was coded both as a cognitive and experiential factor and a health system factor).

## Implication for research and practice

The multitude of factors identified in this review are often inter-related and interacting, and span across personal, family, community and society levels. For example, the association between occupation and healthcare utilisation was reported in several studies.[48 60 66 80 82 104] The effect of predominantly casual work undertaken by slum residents on their healthcare access could be mediated through working hours, income level, knowledge of health and available services, etc. There is also possibility that occupation was associated with health status and hence needs for healthcare services, instead of/in addition to behaviour of using health services. Teasing out the complicated relationships between various determinants and their interaction with the diverse contexts of slums will require in-depth analysis and a more holistic approach to synthesising the evidence. Given the unique features of individual slums, service planners and policy makers will need to examine these relationships with due

consideration to the context specific to each locality and geospatial features and neighbourhood effects that characterise slum settings.[4]

We found far fewer studies that have examined health service providers' perspective than studies that have investigated factors associated with accessing healthcare from slum residents' perspective. There may be scope for greater research and policy attention to supply side factors, including experiences and practices of local frontline healthcare providers, availability of healthcare facilities and infrastructure and policy to support them in order to overcome the many barriers highlighted from both supply and demand sides.

Although only six of the included studies explored factors associated with healthcare access or health service provision across slum and non-slum settings, they showed a generally encouraging picture that access to and provision of healthcare are continuously evolving (and often improving) in slums and other settings, and equality between different settings is not beyond reach.

## CONCLUSION

This scoping review summarises a large body of recent literature evaluating factors associated with seeking and utilisation of healthcare by slum residents, but found substantially fewer studies examining factors associated with provision of health services from providers' perspective. Recent migration into slums; knowledge, perception (including misconception and distrust) and past experience of illness, healthcare needs and health services; financial constraint, competing priorities and inadequacy of social support; adverse physical environment and unfavourable locality; sociocultural expectations and stigma; lack of official recognition; and various problems in existing health system all contribute towards the challenges faced by slum residents. Future research and policy aimed at improving healthcare services in slums should pay more attention to supply side issues ranging from individual healthcare providers and practices to structural and policy-level factors to tackle different barriers faced by slum residents, which in turn need to be evaluated holistically and take into account local context and geospatial features of slums.

**Author affiliations**
[1]Warwick Medical School, University of Warwick, Coventry, UK
[2]KM Data Division, Korea Institute of Oriental Medicine, Daejeon, Republic of Korea
[3]Health and Systems for Health, African Population and Health Research Center, Nairobi, Kenya
[4]Information and Digital Group, University of Warwick, Coventry, UK
[5]Sociology, University of Ibadan, Ibadan, Nigeria
[6]Centre for Health Policy, University of the Witwatersrand, Johannesburg-Braamfontein, South Africa
[7]Population Health Sciences Institute, Newcastle University, Newcastle upon Tyne, UK
[8]Institute of Applied Health Research, University of Birmingham, Birmingham, UK

**Collaborators** The Improving Health in Slums Collaborative: African Population and Health Research Centre (APHRC), Nairobi, Kenya: Pauline Bakibinga, Caroline Kabaria, Ziraba Kasiira, Peter Kibe, Lyagamula Kisia, Catherine Kyobutungi, Nelson Mbaya, Blessing Mberu, Shukri Mohammed, Anne Njeri. Aga Khan University, Karachi, Pakistan: Iqbal Azam, Romaina Iqbal, Ahsana Nazish, Narjis Rizvi. Independent University, Dhaka, Bangladesh: Syed Shifat Ahmed, Nazratun Choudhury, Omar Rahman, Rita Yusuf. Nigerian Academy of Sciences, Lagos, Nigeria: Doyin Odubanjo. University of Ibadan, Ibadan, Nigeria: Motunrayo Ayobola, Olufunke Fayehun, Akinyinka Omigbodun, Mary Osuh, Eme Owoaje, Olalekan Taiwo. University of Birmingham, Birmingham, UK: Richard Lilford, Jo Sartori, Samuel Watson. University of Lancaster, Lancaster, UK: Peter Diggle. University of Warwick, Coventry, UK: Navneet Aujla, João Porto de Albuquerque, Yen-Fu Chen, Paramjit Gill, Frances Griffiths, Bronwyn Harris, Jason Madan, Oyinlola Oyebode, Ji-Eun Park, Simon Smith, Grant Tregonning, Olalekan Uthman, Ria Wilson, Godwin Yeboah.

**Contributors** J-EP, BH, MMA, FG and Y-FC conceptualised the scoping review. J-EP carried out literature searches. J-EP, PK, GY, OO and Y-FC participated in study screening and coding. J-EP and Y-FC performed data charting and drafted the initial manuscript. NA, PG and RJL provided critical input during the drafting of the manuscript. All authors commented on and contributed to the revision of subsequent versions and approved the final version for submission. Y-FC is the guarantor for this article.

**Funding** This research is funded by the NIHR Global Health Research Unit on Improving Health in Slums using UK aid from the UK government to support global health research. MMA gratefully acknowledges the support provided by the Warwick Institute of Advanced Study Global Challenges Research Fund Fellowship (IAS/32013/1914). FG receives funding as South Africa Research Chair in Health Policy and Systems from the National Research Foundation, South Africa. RJL is supported by the NIHR Applied Research Collaboration (ARC) West Midlands, UK. Y-FC is supported by Warwick Evidence, which is a Technology Assessment Review team funded by the NIHR Evidence Synthesis Programme. PG is NIHR Senior Investigator and supported by the NIHR ARC West Midlands. Upon submission, NA had joined the Population Health Sciences Institute, Newcastle University (UK).

**Disclaimer** The views expressed in this publication are those of the author(s) and not necessarily those of the NIHR or the UK Department of Health and Social Care.

**Competing interests** None declared.

**Patient and public involvement** Patients and/or the public were not involved in the design, or conduct, or reporting, or dissemination plans of this research.

**Patient consent for publication** Not required.

**Ethics approval** This realist synthesis included literature that is available in the public domain and did not involve the collection of personal data.

**Provenance and peer review** Not commissioned; externally peer reviewed.

**Data availability statement** All data relevant to the study are included in the article or uploaded as supplementary information.

**ORCID iDs**
Ji-Eun Park http://orcid.org/0000-0002-2932-5373
Peter Kibe http://orcid.org/0000-0002-9027-9054
Godwin Yeboah http://orcid.org/0000-0003-4618-3175
Bronwyn Harris http://orcid.org/0000-0003-4695-008X
Frances Griffiths http://orcid.org/0000-0002-4173-1438
Paramjit Gill http://orcid.org/0000-0001-8756-6813
Richard J Lilford http://orcid.org/0000-0002-0634-984X

Yen-Fu Chen http://orcid.org/0000-0002-9446-2761

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
