## [Reviewer comments · BMJ Open]

ARTICLE DETAILS

TITLE (PROVISIONAL)	Factors associated with accessing and utilisation of health care and provision of health services for residents of slums in low and middle income countries: a scoping review of recent literature
AUTHORS	Park, Ji-Eun; Kibe, Peter; Yeboah, Godwin; Oyebode, Oyinlola; Harris, Bronwyn; Ajisola, Motunrayo; Griffiths, Frances; Aujla, Navneet; Gill, Paramjit; Lilford, RJ; Chen, Yen-Fu

VERSION 1 – REVIEW

REVIEWER	Hunter, Benjamin Kings College, London
REVIEW RETURNED	06-Aug-2021

GENERAL COMMENTS	The value of the manuscript lies in its attempt to bring together a large number of studies across contexts to identify a range of pertinent issues and themes relating to healthcare access for people living in slums. This kind of mapping of evidence has the potential to usefully inform further research and policy. The study appears to have been conducted rigorously and in accordance with many systematic/scoping review practices. The methods are reported clearly and the Discussion and Conclusion sections are generally a fair reflection of the contents of the manuscript, acknowledging the limitations in the authors' approach to the research. There is an issue with one passage that comes in the Discussion and which, in its advocacy for insurance-based financing, seems to come more from the authors than from the findings reported in the manuscript: 'Health insurance is one of the key measures to overcome this barrier [...] policies that improve the uptake and utilisation of health insurance as well as reducing healthcare costs for slum residents need to be considered'. The main issue with the manuscript is that it is far from the kind of comprehensive analysis claimed by the authors. The authors make the case for their study by stating that 'factors associated with healthcare seeking behaviour and healthcare utilisation of slum residents and factors related to the provision of health services in slums have not been systematically examined (with the exception of immunisation services)'. They go on to claim that 'to our best knowledge this scoping review is the first that has comprehensively examined relevant factors across different service areas of health care in slums.' Yet the findings of the review are based on English-language academic literature from a ~four-year period, and the searches did not include looking at relevant organisational websites for grey literature. The result is a review that has retrieved many studies but is nonetheless rather narrow in scope; this greatly undermines any claims that the review is 'comprehensive'.
--

	A second issue is that the findings, while considered and wide-ranging, are at times too superficial for the reader to draw much in the way of insights. For example in the 'Demand side' section the authors' commentary often does not indicate the direction of claimed associations between factors and healthcare access, leaving us guessing what these may be. Throughout the findings there is relatively infrequent attention to the context of slums, and little indication of how the issues being presented may or may not be specific to slums, or whether they are experienced differentially by people living in slums. Closer attention to intersecting issues of class, gender, race and caste might also help to nuance the analysis.
--	--

REVIEWER	Cho, Sung-il Seoul National University Graduate School of Public Health
REVIEW RETURNED	28-Nov-2021

GENERAL COMMENTS	The paper provides an extensive review on the health service utilization in slums, considering it is a "scoping" review. Several points below may be considered in the future revision to enhance its contribution to the literature.  1. It will be much clearer if the actual combinations of search terms and the returned numbers for each step are provided, so that the process match with Figure 3. Please consider adding a table with these results. 2. It seems that search strategy did not include the "health service utilization" components. If so, the authors would have made a large amount of efforts to classify by in this category, restricting from 14041 3895 in Figure 3. The reason for this manual process without using search terms needs to be explained, e.g., lack of appropriate terms or too many missing by any restriction by such terms, etc. 3. Tables 2 & 3 include health system factors which is somewhat unclear about the classification criteria. For example, facility & resources appear in both tables and clear distinction has not been provided. Also, fear of side effects is in Table 3, whereas late opening time is in Table 2, which seem to be as well switched to the other category. A clear theoretical framework for this classification is needed with appropriate reference to previous literature on this conceptualization. 4. Figure 1 & 2 seem to suggest that the theoretical framework has been formulated by previous research before this paper as 1, and then updated to 2 based on the results of the current paper. But it is not clear in the manuscript on what basis each figure has been formulated, especially what aspects of figure 2 has been developed from the results of this paper. The connection would be informative. 5. Concept and definition of "access to health care" is confusing in the manuscript and in the figures 1 & 2. Please provide the definition, especially in relation with WHO health system building block model. In the current paper, access to health care seems to be a mix of population demand (health needs) and actual use (met need), which would already involve provision factors. Clarifying what aspect in the authors' new model improves on what limitations of the older framework of previous literature, and how the improvements
--

	resulted from the current review would be important. 6. Missing affordability issue: It seems to me that in the Figure 1 & 2, affordability issue is almost completely missing. Please clarify how affordability problem in the slums would be reflected in the model and what would be the policy implications.
--	--

VERSION 1 – AUTHOR RESPONSE

Reviewer: **1**
Dr. Benjamin Hunter, Kings College, London

Comments to the Author:
 The value of the manuscript lies in its attempt to bring together a large number of studies across contexts to identify a range of pertinent issues and themes relating to healthcare access for people living in slums. This kind of mapping of evidence has the potential to usefully inform further research and policy.

The study appears to have been conducted rigorously and in accordance with many systematic/scoping review practices. The methods are reported clearly and the Discussion and Conclusion sections are generally a fair reflection of the contents of the manuscript, acknowledging the limitations in the authors' approach to the research. There is an issue with one passage that comes in the Discussion and which, in its advocacy for insurance-based financing, seems to come more from the authors than from the findings reported in the manuscript: 'Health insurance is one of the key measures to overcome this barrier ... policies that improve the uptake and utilisation of health insurance as well as reducing healthcare costs for slum residents need to be considered'.

Thank you for the positive comments on our methodological approach and for pointing out the issue related to the specific text on health insurance. We have revised the passage to ensure that our description is neutral and it reflects and highlights findings from the included studies:

Healthcare cost is a major barrier between the intention to seek care and actual utilisation of services.^{111 139} Health insurance could be one of the potential measures to overcome this barrier.^{151 152} Although possession of/coverage by health insurance was associated with higher levels of utilisation of health services among slum residents,^{18 84} studies showed that uptake of government-run public insurance among slum residents was low.^{70 85} This may be attributed to lack of awareness, difficulties in navigating through the health system and in obtaining official proof of identity required for enrolment,⁷⁰ and poor quality of care and range of services offered.^{70 85} Even among slum residents covered by health insurance, access to care was often refused and additional charges were frequently requested.⁹⁶ Policies that aim to improve access to healthcare services among slum residents through public health insurance will need to address these challenges. (page 29-30)

The main issue with the manuscript is that it is far from the kind of comprehensive analysis claimed by the authors. The authors make the case for their study by stating that 'factors associated with healthcare seeking behaviour and healthcare utilisation of slum residents and factors related to the provision of health services in slums have not been systematically examined (with the exception of immunization services)'. They go on to claim that 'to our best knowledge this scoping review is the first that has comprehensively examined relevant factors across different service areas of health care in slums.' Yet the findings of the review are based on English-language academic literature from a ~four-year period, and the searches did not include looking at relevant organisational websites for

grey literature. The result is a review that has retrieved many studies but is nonetheless rather narrow in scope; this greatly undermines any claims that the review is 'comprehensive'.

We take the peer reviewer's point that our scoping review has several limitations given the restrictions in terms of publishing year and language and the lack of coverage of grey literature. These restrictions needed to be imposed due to the larger-than-expected volume of literature that we found and the resource and time constraint that we had as we stated in the manuscript. In response to the reviewer's comments, we have:

- updated our searches in December 2021 (in response to the editor's comment) and added evidence from recently published papers
- additionally searched organisational websites including Slum Dwellers International, UN HABITAT, UN and WHO, but did not identify relevant studies (we are aware that the search was by no means comprehensive).
- revised various texts to ensure that we do not overclaim the comprehensiveness of this scoping review.

We hope these adequately address the peer reviewer's concerns. As we stated in the manuscript, we are aware of only one published systematic review that focuses specifically on immunisation in slum settings. Therefore, we believe that our scoping review, which is fairly inclusive within the stated constraints, is still a valuable addition to the literature and can highlight the needs for more in-depth synthesis of the rapidly growing number of studies related to accessing, utilisation and provision of health services in slum settings.

A second issue is that the findings, while considered and wide-ranging, are at times too superficial for the reader to draw much in the way of insights. For example in the 'Demand side' section the authors' commentary often does not indicate the direction of claimed associations between factors and healthcare access, leaving us guessing what these may be. Throughout the findings there is relatively infrequent attention to the context of slums, and little indication of how the issues being presented may or may not be specific to slums, or whether they are experienced differentially by people living in slums. Closer attention to intersecting issues of class, gender, race and caste might also help to nuance the analysis.

We appreciate the peer reviewer's comment concerning the depth of analysis. This is an inherent limitation of a scoping review. We have been careful not to claim that we have undertaken in-depth analyses of the data and have repeatedly emphasized the complexity of relationship between different factors, and have added text to highlight the importance of considering the context of individual slums:

"We attempted some preliminary synthesis to configure the identified evidence but have not explored the complex relationship between the factors identified and their interplay with the context of individual slums in depth. Nevertheless, findings from this scoping review will provide a good foundation for further syntheses." (Pages 31)

"Teasing out the complicated relationships between various determinants and their interaction with the diverse contexts of slums will require in-depth analysis and a more holistic approach to synthesising the evidence. Given the unique features of individual slums, service planners and policy makers will need to examine these relationships with due consideration to the context specific to each locality and geospatial features and neighbourhood effects that characterise slum settings."⁴ (Page 33)

We had not indicated the direction of claimed associations because the direction of some associations may vary depending on specific contexts, or may be conflicting between studies. Nevertheless, we agree that where possible some indication of the direction of associations could be helpful for readers, and have now tried to provide this information in relevant tables and some additional descriptions of the specific associations. We acknowledge that more in-depth analyses would generate more insight, but this would be beyond what could be achieved in a scoping review.

Determining whether an issue is unique to slum settings requires a broader synthesis of evidence of literature across both slum and non-slum settings. Considering the diversity in the characteristics of different slums and the complex relationship between the many factors and contexts, such a synthesis is best carried out in reviews with a narrower scope focusing on specific issues and is beyond the scope of our scoping review. Nevertheless, we have included and presented findings from studies that examined data from both slum and non-slum settings, which provide the strongest evidence by making within-study comparisons.

Reviewer: 2
Prof. Sung-il Cho, Seoul National University Graduate School of Public Health

Comments to the Author:
The paper provides and extensive review on the health service utilization in slums, considering it is a "scoping" review. Several points below may be considered in the future revision to enhance its contribution to the literature.

1. It will be much clearer if the actual combinations of search terms and the returned numbers for each step are provided, so that the process match with Figure 3. Please consider adding a table with these results.

Thank you for your comments. We have added the number of searched papers in Figure 3 and provided detailed search strategies in appendix 1. (Figure 3, Appendix 1)

2. It seems that search strategy did not include the "health service utilization" components. If so, the authors would have made a large amount of efforts to classify by in this category, restricting from 14041 3895 in Figure 3. The reason for this manual process without using search terms needs to be explained, e.g., lack of appropriate terms or too many missing by any restriction by such terms, etc.

This review covers healthcare seeking and provision of health services in addition to healthcare utilization. Therefore, use of the terms 'healthcare utilization/ healthcare services utilization' in the search strategy may result in papers related to healthcare seeking and health service provision being missed. In addition, given the numerous terms related to diverse healthcare needs and different health services, it is impractical to include a comprehensive list of all relevant terms in the search

strategy. We therefore used broader (slum-related) keywords to capture all related literature, followed by a manual process to identify and classify relevant papers.

We have also added an explanation regarding the difference between the number of records retrieved and the number of records screened in Figure 3.

3. Tables 2 & 3 include health system factors which is somewhat unclear about the classification criteria. For example, facility & resources appear in both tables and clear distinction has not been provided. Also, fear of side effects is in Table 3, whereas late opening time is in Table 2, which seem to be as well switched to the other category. A clear theoretical framework for this classification is needed with appropriate reference to previous literature on this conceptualization.

Our presentation of different factors in Tables 2 & 3 reflects the conceptual framework that we show in Figure 2, in which we made a distinction between access to health care (which covers being aware of health needs and forming desire for care, seeking care and trying to reach care) from service users' perspective (demand side) and provision of health services from providers' perspective (supply side), and classified different factors inductively. The framework also highlights that the actual utilisation of health care ('met needs' as described in your comments below) requires a match between demand and supply side factors. As factors associated with actual utilisation of healthcare were explored from the demand side perspective in the vast majority of included studies, we presented these factors alongside factors associated with accessing healthcare in Table 2. Table 3 shows factors associated with healthcare services provision, which largely reflect supply side perspective. As the peer reviewer wittingly noticed, some factors such as facility & resources appear in both tables, and this reflects the different perspective taken / different data sources (i.e. service users or providers) in the original studies from which these factors were identified.

For example, the 'fear of side effects' in Table 3 was identified from Odhiambo et al.'s study, in which community health workers described challenges of implementing a population deworming intervention. Therefore it was classified as a factor related to provision of healthcare services in a 'push' rather than 'pull' mode in slums. Likewise, while 'late opening time' arose from organization of services, it was pointed out as a barrier to accessing healthcare services by slum residents. It was therefore categorised as a demand side factor Table 2. We thank the reviewer for identifying this potential lack of clarity, and have now clearly stated the main distinction between Table 2 and Table 3 with regard to perspectives in the titles of the tables.

4. Figure 1 & 2 seem to suggest that the theoretical framework has been formulated by previous research before this paper as 1, and then updated to 2 based on the results of the current paper. But it is not clear in the manuscript on what basis each figure has been formulated, especially what aspects of figure 2 has been developed from the results of this paper. The connection would be informative.

As we described in the Methods section, Figure 1 was developed from assimilation of existing conceptual models related to healthcare access and service delivery. We provided key references for these models:

“The preliminary framework was developed by the review authors based on existing conceptual models related to healthcare access and service delivery³³⁻³⁶”

We have now added some explanation in the Methods section about the addition of ‘cognitive and experiential factors’ which is the key difference between Figures 1 and 2. (page 7-10)

5. Concept and definition of "access to health care" is confusing in the manuscript and in the figures 1 & 2. Please provide the definition, especially in relation with WHO health system building block model. In the current paper, access to health care seems to be a mix of population demand (health needs) and actual use (met need), which would already involve provision factors. Clarifying what aspect in the authors' new model improves on what limitations of the older framework of previous literature, and how the improvements resulted from the current review would be important.

Thank you for your comments. We are aware that the term “access to health care” has been defined and used in many different ways in the literature, and there is no universally accepted definition. We primarily adapted the conceptual model of health care access proposed by Levesque and colleagues,¹ who developed the model after a comprehensive review of prior literature. In Levesque’s paper, they defined access as “the opportunity to reach and obtain appropriate health care services in situations of perceived need for care”, and further expanded the concept to view access as “the possibility to identify healthcare needs, to seek healthcare services, to reach the healthcare resources, to obtain or use health care services, and to actually be offered services appropriate to the needs for care”. We largely retained their stages of access including identification of healthcare needs, seeking healthcare services and reaching healthcare resource from service user’s perspective, but separated out the actual utilisation of health services from our definition of ‘accessing’ during the refinement of our conceptual framework to highlight that healthcare utilisation involves both service users and providers as you pointed out. The WHO health system building block is a higher level conceptual model primarily focusing on health system and organization from service providers’ perspective. Our conceptual model (like the Levesque model) is more service-user centred, but at the same time allows system-level factors and supply side issues to be highlighted. Most importantly, our model facilitates the linkage and mapping between demand side issues and supply side issues through examination of met (service utilisation) and unmet needs (mismatch between the two sides). Although we have not adopted the WHO health system building block for the supply side of our model, the alternative model that we adopted² is also widely used (e.g. by the Cochrane Collaboration) and covers similar domains.

6. Missing affordability issue: It seems to me that in the Figure 1 & 2, affordability issue is almost completely missing. Please clarify how affordability problem in the slums would be reflected in the model and what would be the policy implications.

As you mentioned, affordability is an important barrier for using healthcare. It depends on both the income and wealth of individual residents and families and the amount of fees charged by healthcare providers for the services. The issue of affordability was mainly captured in literature related to the demand side. As shown in Table 2, previous studies have reported ‘inability to afford care’, ‘income’ and ‘wealth’ as factors related to healthcare services utilization, which was classified as socioeconomic factors in our conceptual framework. (Table 2. Socioeconomic factor- income and wealth). Affordability could have been influenced by how health system is organised and financed and therefore be classified as health system factors related to the supply side, but this was not featured in

studies included in the scoping review.

VERSION 2 – REVIEW

REVIEWER	Hunter, Benjamin Kings College, London
REVIEW RETURNED	10-Feb-2022

GENERAL COMMENTS	The revisions have strengthened the manuscript and largely addressed my concerns. A few further (minor) thoughts: P. 2 line 42 – ‘three studies’; line 56 – ‘existing problems in the health system’? P. 29 line 56 – ‘Health insurance could be one of the potential measures to overcome this barrier.’ It still seems strange to devote attention to insurance financing and to not mention alternatives such as tax-based financing which have been used in many countries and which can be more progressive. But if the authors want to stick with their line of commentary around insurance then they might also caveat the sentence with ‘in some settings’.
---

REVIEWER	Cho, Sung-il Seoul National University Graduate School of Public Health
REVIEW RETURNED	30-Jan-2022

GENERAL COMMENTS	The revised manuscript addresses most of the earlier issues. One additional point below may be considered to further improve the paper. The supply side of the access involves universal coverage goals and processes. More clarification would be helpful on how universal coverage strategy is related to this paper's framework.
--

VERSION 2 – AUTHOR RESPONSE

Reviewer: 1
Dr. Benjamin Hunter, Kings College, London

Comments to the Author:
The revisions have strengthened the manuscript and largely addressed my concerns. A few further (minor) thoughts:

P. 2 line 42 – ‘three studies’; line 56 – ‘existing problems in the health system’?
⇒ We edited these as recommended. Thank you.

P. 29 line 56 – ‘Health insurance could be one of the potential measures to overcome this barrier.’ It still seems strange to devote attention to insurance financing and to not mention alternatives such as tax-based financing which have been used in many countries and which can be more progressive. But if the authors want to stick with their line of commentary around insurance then they might also caveat the sentence with ‘in some settings’.
⇒ By ‘health insurance’ we were referring to various types of healthcare financing that do not rely on

out-of-pocket expenditure at the point of accessing care. It was meant to cover both private health insurance and compulsory social health insurance, the latter being closer to tax-based financing model. We agree that the limited evidence included in our scoping review does not provide a strong justification to recommend one model of healthcare financing over the other, and have revised the text in response to your concern and to avoid confusion. (page 29-30)

Reviewer: 2

Prof. Sung-il Cho, Seoul National University Graduate School of Public Health

Comments to the Author:

The revised manuscript addresses most of the earlier issues.

One additional point below may be considered to further improve the paper.

The supply side of the access involves universal coverage goals and processes. More clarification would be helpful on how universal coverage strategy is related to this paper's framework.

⇒ We added some explanation that reducing out-of-pocket expenditure through various forms of healthcare financing could help achieve universal coverage goals, but there is insufficient evidence from studies included in this review to make a firm recommendation. We have also highlighted that initiatives targeting primary care in slums could potentially improve access and universal coverage (page 29-30)

Healthcare provision supported by tax-based financing and/or various forms of social and private insurance that reduce out-of-pocket expenditure at point of care could be potential measures to overcome this barrier and help achieve universal coverage goals. Limited evidence showed that initiatives prioritising primary healthcare coverage in slums could improve access,¹⁴⁵ but there is insufficient evidence from studies included in this review to determine the best model of healthcare financing for improving healthcare access and coverage in slum settings.